# *Streptococcus pyogenes* NAD+-Glycohydrolase Reduces Skeletal Muscle βNAD+ Levels Independently of Streptolysin O

**DOI:** 10.3390/microorganisms10071476

**Published:** 2022-07-21

**Authors:** Eric R. McIndoo, Emily Price, Cheri L. Lamb, Christopher S. Dayton, Clifford R. Bayer, Dennis L. Stevens, Amy E. Bryant, Sarah E. Hobdey

**Affiliations:** 1Infectious Diseases Section, Veterans Affairs Medical Center, Boise, ID 83702, USA; eric.mcindoo@va.gov (E.R.M.); emily.price@va.gov (E.P.); cheri.lambmcfarlane@gmail.com (C.L.L.); topherdayton@u.boisestate.edu (C.S.D.); cbayer2@msn.com (C.R.B.); dennis.stevens@va.gov (D.L.S.); 2Idaho Veterans Research and Education Foundation, Boise, ID 83702, USA; 3Department of Biomedical and Pharmaceutical Sciences, Idaho State University, Meridian, ID 83642, USA; amybryantphd@gmail.com; 4Biomolecular Research Core, Boise State University, Boise, ID 83725, USA; 5Department of Medicine, University of Washington, Seattle, WA 98195, USA

**Keywords:** *Streptococcus pyogenes*, bacterial toxins, pathology, skeletal muscle

## Abstract

Necrotizing soft tissue infections caused by *Streptococcus pyogenes* (group A streptococcus [GAS]) are characterized by rapid and extensive necrosis of fascia and muscle. Molecular epidemiological studies have demonstrated a positive correlation between GAS isolates that cause invasive infections and the production of *S. pyogenes* NAD+-glycohydrolase (SPN), an NADase secreted by GAS, but the effect of SPN on muscle cells has not been described. Thus, using standard βNAD+ and ATP quantification assays, we investigated the effects of SPN on cultured human skeletal muscle cell (SkMC) βNAD+ and ATP with and without streptolysin O (SLO)–a secreted cholesterol-dependent cytolysin known to act synergistically with SPN. We found that culture supernatants from GAS strains producing SLO and SPN depleted intracellular βNAD+ and ATP, while exotoxins from a GAS strain producing SLO and an enzymatically-inactive form of SPN had no effect on βNAD+ or ATP. Addition of purified, enzymatically-active SPN to NADase-negative culture supernatants or sterile media reconstituted βNAD+ depletion but had no effect ATP levels. Further, SPN-mediated βNAD+ depletion could be augmented by SLO or the homologous cholesterol-dependent cytolysin, perfringolysin O (PFO). Remarkably, SPN-mediated βNAD+ depletion was SkMC-specific, as purified SPN had minimal effect on epithelial cell βNAD+. Taken together, this study identifies a previously unrecognized role for SPN as a major disruptor of skeletal muscle βNAD+. Such activity could contribute to the rapid and widespread myonecrosis characteristic of severe GAS soft tissue infections.

## 1. Introduction

Necrotizing soft tissue infections due to *Streptococcus pyogenes* (group A streptococcus [GAS]) remain a significant cause of morbidity and mortality worldwide. Necrotizing infections are characterized by rapid and widespread destruction of fascia, muscle and skin, and they require urgent surgical intervention, including amputation, to ensure survival [1]. The mechanisms of rapidly progressive tissue destruction are not fully understood, though previous studies suggest that the exotoxins streptolysin O (SLO) and *S. pyogenes* NAD glycohydrolase (SPN) play important roles in this process.

Data from our laboratory and others have demonstrated a strong molecular epidemiological association between isolates causing invasive disease and the production of SPN and SLO [2,3]. As a cholesterol-dependent cytolysin, SLO disrupts eukaryotic cell membranes by oligomerizing on the surface of cells and inserting into the membrane to create large pores resulting in cellular cytolysis, apoptosis, and pyroptosis [4,5,6]. Our group has also shown that SLO stimulates formation of intravascular aggregates of platelets and leukocytes that irreversibly block blood flow and contribute to ischemic destruction of muscle tissue [7]. As a potent NADase, SPN is known to mediate the loss of cellular βNAD+ in some eukaryotic cell types (i.e., epithelial cells, keratinocytes [8]) by cleaving βNAD+ to produce nicotinamide and adenosine diphosphoribose. Such activity in skeletal muscle cells has not been examined but could augment the rapid muscle destruction characteristic of severe GAS soft tissue infections. 

Because SLO and SPN work in concert to facilitate toxin entry into some cell types [8] and because binding of SLO to SPN stabilizes both toxins and augments their respective cellular activities [9], the present study investigated the direct effects of SPN and SLO, alone and in combination, on cultured human skeletal muscle cell (SkMC) energetics. The results presented here demonstrate that purified SPN alone can deplete SkMC βNAD+ and that this activity is augmented by SLO, or the homologous cytolysin perfringolysin O (PFO). Furthermore, βNAD+ depletion mediated by purified SPN appeared muscle cell-specific, as minimal activity was observed in SPN-treated epithelial cells. 

## 2. Materials and Methods

*Bacterial strains*-Strain 88-003 is an M-type 3 GAS (M3) recovered from the blood of a patient with Streptococcal toxic shock syndrome (StrepTSS) complicated by myonecrosis. This strain has been characterized in a previous report [10] and has been submitted to the American Type Culture Collection (ATCC #51500). The second, strain 96-004, is an M-type 1 GAS (M1) pharyngitis isolate that is representative of the globally disseminated M1 strains causing invasive disease worldwide [2]. Genetic profiling indicated both strains harbored *nga*, *slo*, *spea* and *speb*. Strains were routinely cultured in Todd Hewitt broth (BD Difco, Franklin Lakes, NJ, USA). 

Laboratory strains included the M-type 5 Manfredo strain (M5), and its SLO-deficient isogenic mutant (M5ΔSLO), obtained from Dr. Michael Kehoe, University of Newcastle upon Tyne, England [11,12]. Neither strain produced detectable NADase activity. 

For preparation of toxin-containing culture supernatants, organisms were cultured in Todd Hewitt broth at 37 °C in 5% CO_2_ with gentle shaking (100 rpm). After 4.5–8 h (late log/early stationary phase cultures), bacteria were removed by centrifugation and the Culture supernatant was filtered through a 0.45 μm low protein binding filter. The resultant material was aliquoted and stored at −70 °C until used for toxin activity assays or cell culture studies. Non-inoculated Todd Hewitt broth was processed in similar fashion and used as a negative control.

*Purification and characterization of extracellular GAS SPN*—Extracellular SPN was purified from an SPEB-deficient mutant of strain M1 as previously described [2]. Briefly, the bacteria-free supernatant from an 18-h culture was concentrated via tangential flow membrane filtration with a 10,000 MW cutoff (Millipore, Burlington, MA, USA). The concentrated material was passed over a CM-Sepharose column equilibrated and washed with 25 mM MES, pH 6.0. SPN was eluted with 25 mM MES pH 6.0 containing 750 mM NaCl. Fractions positive for NADase activity were pooled, dialyzed against deionized water, and subjected to isoelectric focusing in Ultradex gel with a pH gradient of 3.5–9.5. The fractions containing the highest NADase activity were eluted from the gel with 25 mM potassium phosphate buffer pH 7.5 and the carrier ampholytes were removed on a Sephadex G-25 column equilibrated in the same buffer. The resultant material was further purified utilizing a hydroxyapatite column (CHT-II Econo-Pak Cartridge, 5 mL, BioRad, Hercules, CA, USA) equilibrated in 25 mM potassium phosphate buffer pH 7.0. Proteins were eluted ith a potassium phosphate buffer gradient from 0–530 mM at a flow rate of 0.5 mL/min. 

NADase activity of the purified SPN was quantified using an indirect spectrophotometric assay as previously described [2]. Briefly, samples (0.2 mL) were added to 0.8 mL of 100 mM potassium phosphate buffer pH 7.35. The reaction was initiated by addition of 0.2 mL of 7.5 mM βNAD+ (substrate). After 20 min at 37 °C, the reaction was stopped by adding 0.3 mL of 3 N trichloroacetic acid to precipitate the enzyme. Remaining βNAD+ was then quantitated in a coupled enzymatic reaction with alcohol dehydrogenase that reduces βNAD+ to NADH, and absorbance was measured at 340 nm. One unit of NADase activity is defined as the quantity of enzyme that hydrolyzes 1.0 μmole of βNAD/min at pH 7.35 at 37 °C.

Purified SPN showed a single band at 50 kDa by SDS-PAGE and silver staining. Typical preparations contained 600 units of NADase activity/mg of protein. In addition, the purified material had no detectable streptolysin O activity as determined by standard red blood cell hemolysis assay SLO, previously described [13,14], or by Western immunoblot using a monoclonal antibody against SLO (clone 3H10, [15]).

*Recombinant toxins and anti-toxin antibodies*—Recombinant streptolysin O (rSLO) was kindly provided by Dr. Michael Kehoe, University of Newcastle upon Tyne, UK [16]. Recombinant perfringolysin O (rPFO) was generously provided by Dr. Rodney Tweten, University of Oklahoma Health Science Center, Oklahoma City, OK, USA [17]. Hemolytic activities of rSLO and rPFO were determined by sheep red blood cell hemolysis assay in the absence of cysteine as previously described [14,18]. Neutralizing monoclonal antibody against SLO (clone 3H10, a murine IgG_1_) was provided by Dr. Hiroko Sato, formerly of the National Institute of Health, Tokyo, Japan [15] and is now maintained by the Nation Bio-resource Project (Riken BRC, Tokyo, Japan). Polyclonal antibody against streptococcal SPN was provided by Dr. Dieter Gerlach [19] or commercially (Abcam, Cambridge, MA, USA).

*Human cell cultures*—Human primary skeletal muscle cells (Lonza, Basel, Switzerland) were maintained in tissue culture using skeletal muscle growth media (SkGM; Lonza) supplemented with 2.5% fetal bovine serum (Hyclone, Logan, UT, USA) according to the manufacturer’s recommendations. For use in assays, myoblasts of passages 2–4 were seeded at 5 × 10^3^ cells/cm^2^ in 96 well plates and used 1–2 days after achieving confluency (~1 × 10^5^ cells/well). At this time, cells began demonstrating morphologic and genetic changes consistent with early myofibrils. Human epithelial cells (A549s) were maintained in tissue culture using F-12K Medium (Gibco, Gaithersburg, MD, USA) supplemented with 10% fetal bovine serum (Gibco). For use in assays, cells were seeded at 5 × 10^3^ cells/cm^2^ in 96 well plates and used within 3–4 days after achieving ~95% confluency. Hemolytic assays verified that neither concentration of fetal bovine serum protected against SLO-induced lysis.

One day prior to experimentation, confluent cultures of skeletal muscle cells or A549s in 96 well plates were fed with 150 μL fresh SkGM or F-12K medium, respectively. For conciseness, these cell-specific media formulations are subsequently referred to as eukaryotic cell culture media (ECCM). The following day, 100 μL of ECCM was removed and replaced with 50 μL of: (1) PBS or Todd Hewitt broth, as vehicle controls; (2) sterile, non-concentrated GAS culture supernatant diluted two-fold from 1:2–1:128; or (3) purified SPN (0.03125–0.5 units) or rSLO (0.0004–1.5 hemolytic units, HU) alone or in combination. After 0.5, 1, 2, or 4 h at 37 °C in 5% CO_2_, the reaction was terminated, and cellular βNAD+ and ATP were extracted and quantitated as described below. These concentrations of culture supernatant, purified SPN and rSLO were determined in preliminary experiments to be non-cytotoxic cells based on trypan blue exclusion and on normal cellular morphology as visualized by phase contrast microscopy. 

*Extraction and quantitation of cellular ATP*—At designated times, the culture media in each well was removed by gentle aspiration, and the cells were washed once with 100 μL room-temperature Hank’s balanced salt solution (HBSS; Sigma, St. Louis, MO, USA). Next, 100 μL ice-cold ATP extraction buffer (100 mM Tris-acetate pH 7.75 containing 0.25% dodecyltrimethylammonium bromide [DTAB] and 2 mM EDTA) was added, and the plate incubated on ice for 10 min, as described in [20]. The plate was spun at 300× *g* at 4 °C for 2 min, and 75 μL of supernatant from each well was transferred to individual microcentrifuge tubes and kept on ice. ATP (Sigma) was resuspended in ATP extraction buffer to 2 mg/mL and serially diluted to create a standard curve. 50 µL of each standard or cellular extract was placed in duplicate wells of the 96 well plate. Next, 110 μL of 100 mM Tris-acetate buffer pH 7.75 without DTAB or EDTA was added to each well and the plate placed in a luminometer (Fluoroskan Ascent FL, Thermo Scientific, Waltham, MA, USA) at 37 °C. Firefly luciferin/luciferase extract (Sigma) was reconstituted in deionized water according to the manufacturer’s instructions and insoluble material was removed by low-speed centrifugation. Luminescence was read immediately following automatic injection of 40 μL firefly luciferin/luciferase substrate and every second thereafter for 10 s. The concentration of ATP in test samples was extrapolated from a curve relating standard amounts of ATP in ng/mL to peak luminescence values. The assay was linear over the range of 0.1–20 μg/mL with a lower level of sensitivity of 20 ng/mL ATP.

*Extraction and quantitation of cellular βNAD+*—At selected times, the culture media ± SPN/SLO was aspirated from each well and the wells washed once with HBSS. To prevent hydrolysis of βNAD+ that is released upon cell lysis by any residual SPN remaining in the well, 300 μL of rabbit polyclonal anti-SPN (ProSci, Fort Collins, CO, USA) containing high titer neutralizing activity against SPN was diluted 1:20 in PBS and added to the samples and incubated for 15 min at 37 °C (~20 neutralizing units per mL). As a control for neutralization of residual SPN, 0.5 U SPN/well was added to cells for ~30 s, aspirated and diluted anti-NADase serum was added and incubated as for experimental wells. After neutralization, anti-NADase serum was aspirated and 100 μL of 0.4 M perchloric acid was added for 20 min at room temperature to extract βNAD+ [21]. Extracts were neutralized empirically with 2.5 M potassium bicarbonate and precipitates were removed by centrifugation. 10 μL samples were transferred to wells of a 96 well plate for quantification of βNAD+. 

The concentration of βNAD+ in skeletal muscle cell extracts was measured using a cycling assay [22]. With this method, βNAD+ present in the sample is reduced to NADH by alcohol dehydrogenase. NADH then reduces phenazine ethosulfate which in turn reduces a terminal acceptor, MTT, (3-[4,5-dimethylthiazol-2-yl]-2,5-diphenyltetrazolium bromide) causing a precipitate to form and regenerating oxidized phenazine ethosulfate and βNAD+, starting the cycle anew. Reduced MTT is then measured by absorbance at 550 nm. Thus, the concentration of βNAD+ in a given sample is directly related to the time-dependent reduction of MTT [22]. To conduct this assay, 250 μL of freshly prepared βNAD+ assay reagent (2.5 mM phenazine ethosulfate, 0.5 mM MTT, 3.3% ethanol, and 0.25 mg/mL alcohol dehydrogenase in 50 mM bicine, 50 mM nicotinamide pH 7.6) was added to duplicate wells of the prepared cell extracts. After 30 min, absorbance was measured at 550 nm on an ELISA plate reader. The concentration of βNAD+ in the cell extracts was obtained using a standard curve relating absorbance values to known βNAD+ concentrations (0.125–2.0 μg/mL). 

*Data fitting and statistics*—All data were analyzed using GraphPad Prism 9. The 50% effective dilution (EC50) ± the 95% confidence interval (CI) was determined using non-linear regression Y = B + X ∗ (T − B)/(ED50 + X), where B and T are the bottom or top of the plateaus, respectively, ED50 is the 50% effective dose and B = 0 to model complete βNAD+ depletion. The 50% effective concentrations (EC50) ± the 95% CI were determined using non-linear regression Y = B + (T − B)/(1 + (X/EC50)) where B and T are the bottom or top of the plateaus, respectively, EC50 is the 50% effective concentration and B = 0 to model complete βNAD+ depletion. Curves were compared using extra sum-of-squares F test to test whether the data were different from each other, and P values were reported. For grouped data, analysis of statistical differences between groups was completed using two-way ANOVA and the indicated multiple comparison tests post hoc. *p* ≤ 0.05 was considered significant. All biological replicates were independent experiments completed on different days with fresh preparation of experimental components. Each biological replicate had 2–3 technical replicates, which were wells seeded on the same day, from the same bulk culture and treated with same preparation of all components. The averages of the technical replicates were used to determine the value of a single biological replicate.

## 3. Results

*GAS exotoxins reduce key energy intermediates in human SkMCs*—Cultured SkMCs were treated with late log/early stationary phase M-type 1 (M1) or M-type (M3) GAS culture supernatants. Non-concentrated M1 and M3 culture supernatants contained 2.13 and 1.63 NADase Units (U)/mL and 2 and 8 hemolytic units (HU)/mL of SLO, respectively. Both M1 and M3 culture supernatants caused dose-dependent depletion of βNAD+ and ATP in SkMCs (Figure 1A,B) with maximal depletion of βNAD+ and ATP occurring at 1- and 2-h post-toxin exposure, respectively (data not shown). A 50% reduction of βNAD+ was achieved with a 1:20 dilution of the M1 culture supernatant, whereas a 1:10 dilution of the M3 culture supernatant was necessary for the same effect (Figure 1A). In contrast, the 50% reduction of ATP was completely overlapped between the two culture supernatants (Figure 1B). None of these toxin exposures resulted in significant cell cytotoxicity or cell loss (Appendix A).

*SPN reduces intracellular βNAD+ in human skeletal muscle cells*—SPN mediates the intracellular loss of βNAD+ in some eukaryotic cell types [8], though nothing has been reported about this activity in myocytes. Such depletion of this key energy intermediate in skeletal muscle cells could contribute to the widespread destruction of muscle tissue characteristic of GAS myonecrosis. To test whether SPN alone is responsible for culture supernatant-induced loss of intracellular βNAD+ and/or ATP in human myocytes, two studies were undertaken. First, human SkMCs were treated with culture supernatant from a wild-type (WT) M-type 5 (M5) GAS strain that is naturally SLO-positive but produces enzymatically-inactive SPN (SLO^pos^/SPN^inactive^) [23,24,25] or its SLO-deficient isogenic mutant (ΔSLO; SLO^neg^/SPN^inactive^). Results show that exogenous addition of enzymatically active, purified SPN to either culture supernatants or non-inoculated media resulted in a loss of βNAD+, especially in the WT SLO-positive culture supernatant (compare closed to open bars in Figure 2A). In contrast, addition of enzymatically-active SPN caused no changes in ATP levels (compare closed to open bars in Figure 2B). These results illustrate that SPN with or without SLO from culture supernatants can deplete intracellular βNAD+ but not ATP and suggest that SLO augments SPN-mediated βNAD+ depletion in SkMCs.

Next, to confirm SPN’s independent role in reducing intracellular βNAD+, a dose-response study was completed using only purified SPN. Similar to findings in Figure 2A,B, results show that purified SPN dose-dependently depleted SkMC βNAD+ with a 50% effective concentration (EC50) of 0.05 U/well (Figure 2C, closed circles) but had no effect on SkMC ATP (Figure 2C, open circles). Since SPN was natively purified, contaminating SLO was ruled out in a separate experiment in which treatment of purified SPN with neutralizing anti-SLO antibody did not alter SPN’s ability to reduce cellular βNAD+ (Appendix A). To verify that intracellular β-NAD depletion was not due to leakage from damaged cell membranes, experiments were repeated in the presence of trypan blue to image cells with compromised membranes by light microscopy (Appendix A). We found that none of the concentrations of SPN used caused membrane damage. These findings demonstrate that SPN alone reduces intracellular βNAD+ in human SkMCs and suggest that SPN is not depleting βNAD+ by damaging the cell membrane.

*SPN-induced depletion of βNAD+ in SkMCs is augmented by SLO*—In addition to being a potent cytolysin, SLO also mediates transport of SPN into epithelial cells and keratinocytes [8]. Thus, we sought to determine if SPN activity in skeletal muscles is also augmented by SLO using two approaches. First, 8-fold dilutions of culture supernatant from WT M1 and M3 GAS were pretreated with excess neutralizing anti-SLO antibody [15] or an irrelevant isotype-matched control antibody (murine IgG1) before addition to cultured SkMCs. Comparable to data presented in Figure 2A, results show substantial depletion of cellular βNAD+ in the presence of M1 or M3 culture supernatants and clearly illustrate that eliminating SLO activity significantly reduced the amount of βNAD+ depletion (Figure 3A). While the difference did not reach statistical significance, SLO neutralization also afforded some protection against ATP depletion compared to vehicle or IgG isotype controls (Figure 3B).

Second, to investigate whether SPN activity is enhanced by SLO in the absence of other exotoxins present in culture supernatants, recombinant SLO (rSLO) and purified SPN were tested for their individual and combined effects on SkMC βNAD+ and ATP levels. In the absence of SPN, rSLO alone caused only low level βNAD+ depletion (Figure 3C, closed squares). In contrast, depletion of βNAD+ was clearly increased in SkMCs treated with rSLO plus SPN (Figure 3C, open squares). No depletion of intracellular ATP was observed in SkMCs treated with rSLO or SPN alone or in combination (Figure 3B); in fact, a small, but insignificant increase in ATP was observed with low level rSLO (+/− SPN). Together these findings suggest that SLO significantly augments SPN-mediated βNAD+ depletion while having only a minor effect in ATP depletion.

*Cytolysin-mediated augmentation of βNAD+ depletion is not SLO-restricted*—Our finding that SLO facilitates SPN intracellular activity in skeletal muscle cells is consistent with previous findings that SLO mediates translocation of SPN into epithelial cells and keratinocytes [8]. This same group has also shown that translocation of SPN into these cell types is SLO-specific and cannot be accomplished by other cholesterol-dependent cytolysins [26], but it is not known whether this holds true in SkMCs.

The cholesterol-dependent cytolysin family includes SLO from *S. pyogenes*, perfringolysin O (PFO) from *Clostridium perfringens* and pneumolysin (PLY) from *Streptococcus pneumoniae*, among others. SLO and PFO share conserved amino acids required for cholesterol binding and hemolytic activity [27] and both contribute via similar mechanisms to the pathogenesis of their respective necrotizing soft tissue infections. Thus, we hypothesized that, unlike keratinocytes and epithelial cells, PFO might substitute for SLO in the transport of SPN into skeletal muscles. Human SkMCs were exposed to sub-cytotoxic doses of rPFO, alone or in combination with 0.03 U of purified SPN. Like rSLO, rPFO alone caused low-level loss of βNAD+ (Figure 4, close triangles) and also augmented the ability of SPN to reduce intracellular βNAD+ in human skeletal muscles (Figure 4, open triangles).

*SPN-induced βNAD+ depletion is cell-type specific*—While SLO facilitates SPN intracellular activity our data suggest that SPN alone is sufficient to reduce βNAD+ in SkMCs. To confirm the apparently unique responses of SkMCs to SPN, we repeated our assays with cultured human alveolar basal epithelial cells (A549s) using conditions identical to those for SkMCs. Results show some loss of intracellular βNAD+ from epithelial cells with increasing concentrations of purified SPN (Figure 5A), but the EC50 was 6-fold higher for A549s than SkMCs (EC50 = 0.3 vs 0.05 U/well for A549s and SkMCs, respectively). Further, rSLO had no singular effect on βNAD+ in A549 cells and, SLO at all doses tested (0.001–1.5 HU), failed to augment NADase activity in these cells (Figure 5B, open squares vs open circles). Notably, no cell lysis was observed in A549 cells at the highest SLO dose tested, determined by trypan blue staining and microscopic inspection (Appendix A).

## 4. Discussion

Rapid destruction of viable healthy muscle tissue is a hallmark feature of invasive soft tissue infection caused by GAS. Clinically, myonecrosis in GAS infection is suspected in patients with creatinine phosphokinase elevation, myoglobulinemia, or severe muscle pain, and is substantiated surgically by an absence of muscle contractility and necrosis of myofibrils [1]. Although no single toxin is known to be responsible for all the damage to the musculature, we have found that SLO is a major player in thwarting the tissue inflammatory response [7] and in causing vascular dysfunction that contributes to ischemic destruction of muscle tissue [7]. New findings presented here further indicate that SPN, alone or in combination with SLO, significantly and specifically depletes a key skeletal muscle energy intermediate that could contribute to rapid muscle destruction characteristic of GAS myonecrosis. 

The concept of functional synergy between SLO and SPN is not new. In 2001, Madden et al. first described cytolysin-mediated translocation (CMT) in which SLO facilitated translocation of SPN from host cell-adherent GAS into the cytoplasm of human epithelial cells (A549) and keratinocytes [8] (Figure 6A). Results from their study also illustrated that CMT did not occur during mixed infection with GAS strains producing either SLO or SPN alone, implying that the process of translocation depended on a specific interaction between the two proteins prior to or during secretion. This same group subsequently showed that CMT was pore-formation independent [28] and that SPN translocation could not be completed by the highly homologous cholesterol-dependent cytolysin, PFO [26]. While our data do not rule out the possibility that SLO-facilitated transport may be more efficient within the microenvironment of the bacterial-eukaryotic cell interaction, we clearly demonstrate that such an intimate interaction is not required for SPN to profoundly deplete skeletal muscle βNAD+, nor is SLO per se required to augment βNAD+ depletion, since PFO substituted equally well.

Thus, early in GAS myonecrosis, initial tissue destruction is likely a consequence of the unique individual actions of SLO and SPN, and, later in the disease course, rapidly progressive necrosis occurs when toxin concentrations reach levels that facilitate toxin-toxin interactions and functional synergy. At this stage, secreted toxins could easily spread between muscle fibers, killing cells as they advance. Indeed, histologic evidence supports a multimodal mechanism of SLO and SPN activity, as human autopsy specimens demonstrate uniform, widespread destruction of large muscle groups rather than spotty damage to only those myocytes having adherent GAS with localized co-production of SLO/SPN (reviewed in [1]). Histopathologic findings in GAS myonecrosis recapitulate those observed in *Clostridium perfringens* myonecrosis where PFO and the alpha toxin (a phospholipase C) cause widespread destruction of healthy tissue far from the nidus of infection [32], yet more in vivo work is needed to thoroughly assess the toxin profiles in areas of adjacent muscle groups that are devoid of visible GAS. 

Mechanisms by which SPN may contribute to the pathogenesis of invasive streptococcal disease include the loss of βNAD+ [2], production of ADP-ribose [2,33] and enhancement of SLO activity [34]. We also hypothesized that βNAD+ depletion would uncouple electron transport, resulting in a decrease of ATP and ultimately induction of apoptosis. Instead, we found that SPN had no effect on ATP levels in SkMCs, even after four hours of treatment. This finding is in direct contrast to that observed in keratinocytes where SPN depleted cells of both βNAD+ and ATP [34]. While more work is needed to understand the mechanisms involved, it is possible that βNAD+ depletion did not affect SkMC ATP levels due to their ability to produce ATP from large stores of creatine phosphate independently of βNAD+ [35]. Interestingly, and in contrast to purified toxins, culture supernatants reduced skeletal muscle ATP within two hours of treatment, suggesting that another factor in the culture supernatant may contribute to loss of ATP in these cells. However, more experiments are needed to test this hypothesis. 

The fact that SPN depleted SkMC βNAD+ independent of SLO is a new finding. Consequently, the mechanism of SLO-independent SPN translocation can only be speculated. However, a mechanism of SPN- membrane binding has been recently described [29]. In a model of ex vivo infection, SPN bound to an unknown membrane receptor through a putative carbohydrate binding site found in SPN’s N-terminus [29]. The authors propose that this mechanism may be used to orient co-secreted SLO to the membrane for CMT in the absence of cholesterol. Thus, it is not unreasonable to envision that upon membrane carbohydrate binding, SPN may be translocated into the cell independently of SLO, as depicted in Figure 6B. Moreover, carbohydrate profiles vary widely among cell-types, which may partially explain the differences observed here in SPN-mediated βNAD+ depletion between skeletal muscles and epithelial cells, and between epithelial cells and keratinocytes as observed by others [8]. As further evidence that SPN affects specific cell types differently, SPN is a mediator of tissue tropism. Specifically, the ability of GAS to cause both pharyngeal and cutaneous infections correlates with strains producing enzymatically-active SPN, while the ability to cause infection of only the pharynx *or* skin is associated with GAS strains producing enzymatically-inactive SPN [36]. Finally, mutations to the carbohydrate binding site of SPN enhance GAS internalization into cervical epithelial cells (HeLa) but not into gingival epithelial cells (Ca9-22) in a caveolin 1-dependent manner [37]. These studies and ours support that SPN may have distinct roles in mild (epithelial cell involvement) and severe (muscle, adipose, dermal tissue involvement) GAS infections. 

In summary, our findings demonstrate that SPN uniquely and profoundly depletes skeletal muscle βNAD+, that this activity is not dependent on SPN/SLO or GAS/host cell interactions and that this process can be augmented by SLO and the homologous pore-forming toxin PFO. Toxin-mediated disruption of skeletal muscle βNAD+ could dramatically alter the principal function of the myocyte, i.e., contraction, and compromise myocyte survival. This may explain, in part, the temporal association between the emergence of severe GAS infections, including myonecrosis, and production of SPN among associated strains [2].

## Figures and Tables

**Figure 1 microorganisms-10-01476-f001:**
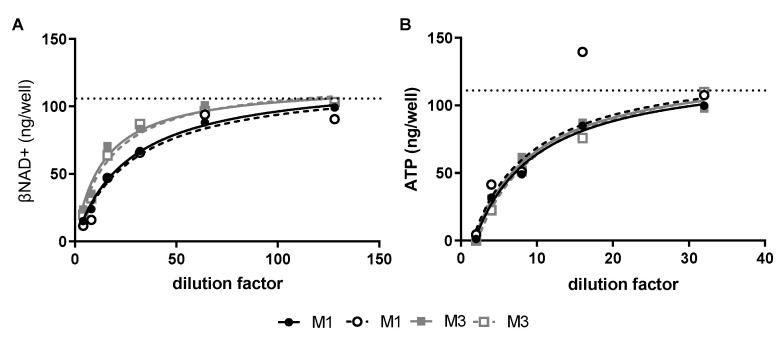
Exotoxins from GAS culture supernatants deplete skeletal muscle βNAD+ and ATP. Graphical representation of intracellular βNAD+ (**A**) or ATP (**B**) from cultured human SkMCs (~1 × 10^5^ cells) after exposure to increasing dilutions of log-phase M1 or M3 GAS culture supernatants. Non-inoculated broth was used as a vehicle control (dashed line). In all graphs, data were fit by non-linear regression to find the 50% effective dilution (ED50) for each data set, i.e., two dilution series for M1 (closed or open circles with black solid or dashed lines that represented the curve fit) and two dilution series for M3 (closed or open squares with grey solid or dashed lines that represented the curve fit). Symbols represent each individual value. (**A**) Intracellular βNAD+ was measured by cycling assay after 1 h of exposure to culture supernatant. For M1, ED50s are 26 and 28 for solid and dashed lines, respectively. For M3, ED50s are 14 and 16 for solid and dash lines, respectively. (**B**) ATP was quantitated by firefly luciferin/luciferase luminescence assay after 2 h of exposure to culture supernatant. For M1, ED50s are 6 and 8 for solid and dashed lines, respectively. For M3, ED50s are 3 and 12 for solid and dash lines, respectively.

**Figure 2 microorganisms-10-01476-f002:**
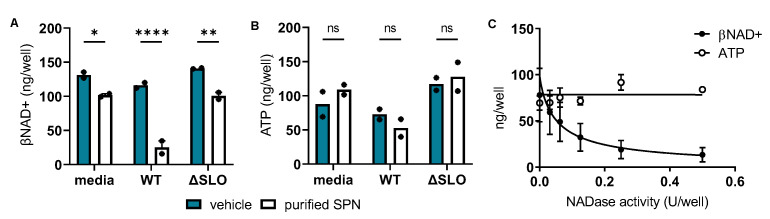
SPN depletes SkMC βNAD+. (**A**,**B**) Cultured SkMCs were treated with an 8-fold dilution of media (Todd Hewitt broth) or culture supernatants from either wild-type (WT) M5 GAS (SLO^pos^/SPN^inactive^) or the SLO-deficient (ΔSLO) isogenic mutant (SLO^neg^/SPN^inactive^) in the absence (closed bars) or presence (open bars) of purified, enzymatically-active SPN (0.03 U). Bars represent the means of n = 2 independent experiments shown as symbols Significance was determined by ANOVA followed by Sidak’s multiple comparison * *p* ≤ 0.02; ** *p* ≤ 0.005; **** *p* ≤ 0.0001; ns = not significant. (**C**) Cultured human SkMCs were exposed to increasing concentrations of purified SPN specified in NADase units per well (U/well). Symbols represent the means of n = 3 (βNAD+) or n = 2 (ATP) independent experiments. Error bars represent SEM. The 50% effective concentration (EC50) ± 95% CI was determined by non-linear regression. EC50 = 0.05 ± 0.05 U/well for βNAD+. EC50 was ambiguous for ATP data.

**Figure 3 microorganisms-10-01476-f003:**
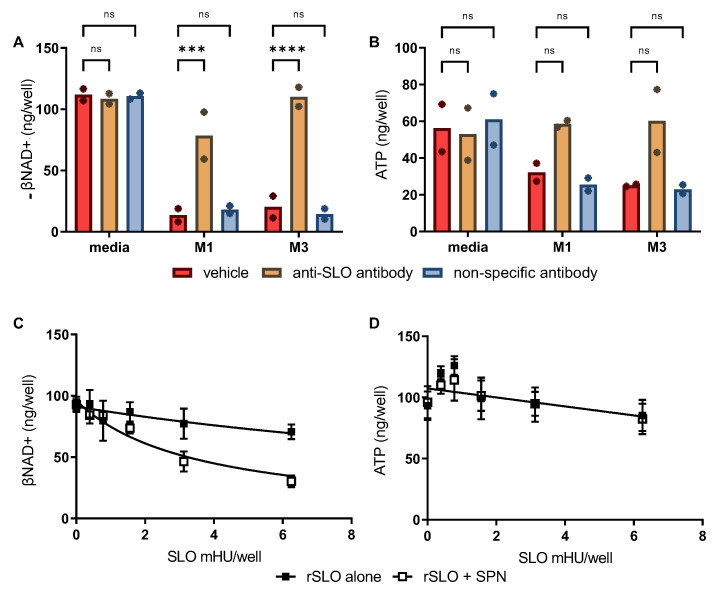
SLO facilitates SPN-mediated depletion of βNAD+. (**A**,**B**) diluted (8-fold) M1 or M3 culture supernatant was incubated at 37 °C for 15 min with neutralizing anti-SLO antibody or an isotype-matched control antibody prior to their addition to cultured skeletal muscle cells. Non-inoculated culture media served as controls. Bars represent the means from an n = 2 independent experiments shown as symbols. Significance was determined by ANOVA. Dunnett’s multiple comparison test was used to compare treatments to vehicle control post hoc. *** *p* ≤ 0.001; **** *p* < 0.0001; ns = not significant. (**C**,**D**) Recombinant SLO (rSLO) alone, or in combination with purified SPN (0.03 U), was added to SkMCs. SLO concentrations are given as milli-hemolytic units (mHU)/well. Symbols represent the means from n = 4 independent experiments. Error bars represent SEM. EC50 ± 95% CI was determined by non-linear regression. (**C**) EC50 = 3.5 ± 0.9 mHU/well for SLO + 0.03 U SPN (open squares) and EC50 = 19.9 ± 14.2 mHU/well for SLO alone. (**D**) Near identical linear data is not reported.

**Figure 4 microorganisms-10-01476-f004:**
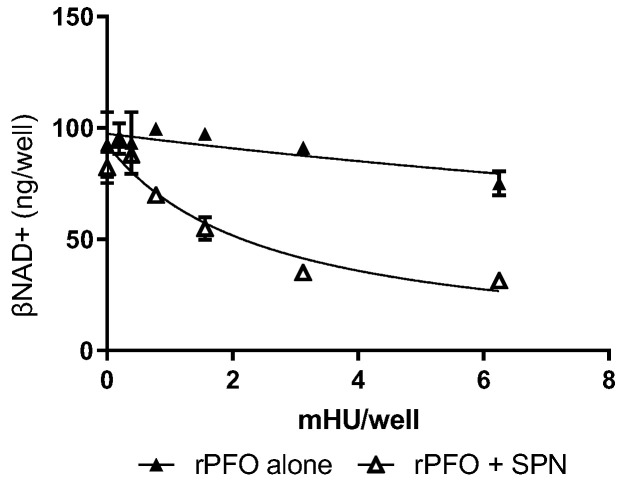
Cytolysin-mediated depletion of βNAD+ is not SLO-restricted. Cultured SkMCs were exposed to rPFO at various mHU/well either alone or in combination with purified SPN (0.03 U). Symbols represent the mean of n = 2 independent experiments. Error bars represent SEM. EC50 ± 95% CI was determined by non-linear regression. EC50 = 2.5 ± 1.0 mHU for PFO + 0.03 U SPN (open triangles) and EC50 = 28 ± 24 for PFO alone (close triangles).

**Figure 5 microorganisms-10-01476-f005:**
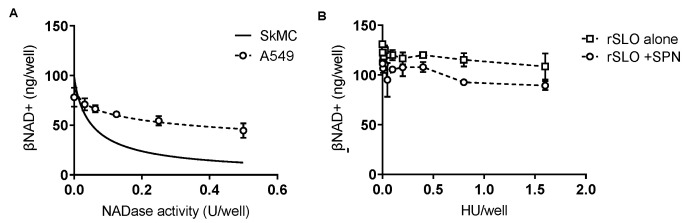
SPN-mediated reduction in intracellular βNAD+ is cell type specific. (**A**) Cultured human epithelial cells (A549s) were exposed to increasing concentrations of SPN, specified in NADase U/well (dotted line). Solid line is for comparison of SkMCs data from Figure 2C. EC50 ± 95% CI = 0.3 ± 0.4 U/well for A549 βNAD+. EC50 for SkMCs is 0.05 ± 0.05 U/well, as reported in Figure 2C. (**B**) A549s were treated with SLO alone or SLO with 0.03U SPN, as done in Figure 3. In all graphs, symbols represent the mean of n = 2 independent experiments. Error bars represent SEM. Linear data not fit.

**Figure 6 microorganisms-10-01476-f006:**
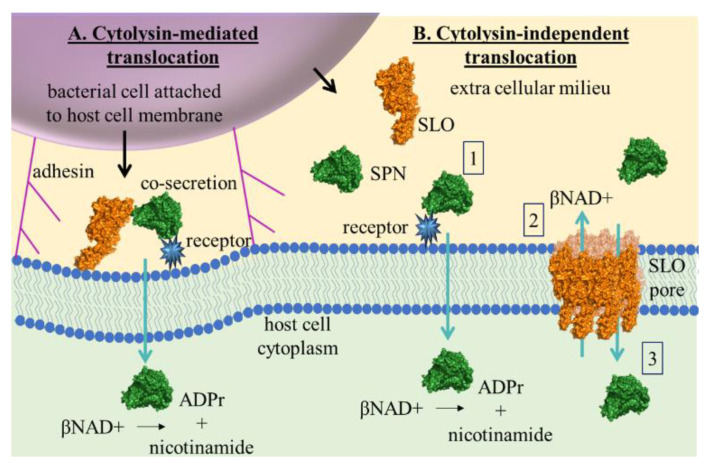
Models of SPN and SLO-mediated βNAD+ depletion. (**A**) Cytolysin-mediated translocation (CMT [8,29]). Co-secreted SPN (green structure, PDB 3pnt [30]) and SLO (orange structure, PDB 4hcs [31], orange) by GAS attached to host cell membrane. SPN binds to unknown carbohydrate receptor (blue star) to orient SLO for cytolysin-mediated translocation. (**B**) Cytolysin-independent translocation (this study). Possible mechanisms include: (1) SPN binds unknown carbohydrate receptor for SLO-independent translocation. SLO augments βNAD+ depletion by (2) forming pores that enable diffusion of βNAD+ out of the cell and/or (3) by facilitating diffusion of SPN into the cell.

## Data Availability

Data will be made available upon email request to the corresponding author.

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
