# Peer review of "Streptococcus pyogenes NAD+-Glycohydrolase Reduces Skeletal Muscle βNAD+ Levels Independently of Streptolysin O"

_microorganisms, 2022, doi:10.3390/microorganisms10071476_

Round 1

Reviewer 1 Report

Suggestions:

Figure 1. Graphical representation of intracellular βNAD”+” A)…

Figure 3. SLO facilitates SPN-mediated depletion of βNAD+ and ATP. Suggestion: to remove “and ATP”.

Author Response

Figure 1. Graphical representation of intracellular βNAD”+” A)…

Modification made.

Figure 3. SLO facilitates SPN-mediated depletion of βNAD+ and ATP. Suggestion: to remove “and ATP”.

Modification Made

Reviewer 2 Report

A well-written, easy-to-follow (revised) manuscript with interesting findings. This reviewer has no issues or comments.

Author Response

A well-written, easy-to-follow (revised) manuscript with interesting findings. This reviewer has no issues or comments.

We thank the reviewer for their positive comments.

This manuscript is a resubmission of an earlier submission. The following is a list of the peer review reports and author responses from that submission.

Round 1

Reviewer 1 Report

In this manuscript, the authors examine the effects of the NAD+ glycohydrolase SPN and the hemolysin streptolysin O (SLO) from Streptococcus pyogenes on intracellular NAD+ and ATP levels in a skeletal muscle cell line.

Although the manuscript is generally technically sound and interesting, there is a major problem with the biological replicates. The data in Figure 1 do not indicate the number of biological replicates on which the data points are based. The plots and statistics in Figures 2 and 3 are mostly based on only n=2 biological replicates, which usually does not allow for statistical calculations. Instead, technical replicates were included. However, this is not permissible for calculations of significance of a biological effect because it artificially increases the sample size. Please include at least 3 biological replicates for graphs and statistical analyses.

Reviewer 2 Report

The conclusion is not supported by the experimental results: Results showed that SPN treatments, with or without SLO, depletes ß-NAD+ but had very minor effects on cell ATP. Therefore, these results did not support the conclusion “SPN disrupts skeletal muscle energetics”. Also, it is clear that SPN depletes more ß-NAD+ in the presence of SLO and SLO seems to be essential for cellular ATP depletion (Fig. 2C). Why SPN disrupts skeletal muscle energetics “independently” of streptolysin O?

In Fig. 2C, the purified SPN depleted intracellular ß-NAD+ but had no effects on intracellular ATP, suggesting that SLO would be required for depleting cellular ATP in the presence with SPN. However, in Fig. 3B, neutralizing SLO by anti-SLO antibody in the supernatants from M1 and M3 strains did not have a significant effect on preventing cellular ATP depletion. These results are not consistent. 

SPN has been known as the cotoxin of SLO (O'Seaghdhast al. 2013. PLoS Pathog 9:e1003394. https://doi.org/10.1371/journal.ppat.1003394); therefore, the novelty of this study is limited. This study showed that SPN would act in a cell type-dependent way; however, to compare SkMC and A549 side-by-side (under the same concentration of SPN) might not be proper. More detailed investigations are still needed to support this conclusion. 

Line 260-262: It is not clear why the LDH assay was performed but did not (or cannot) provide information about the level of cell damage. Also, results from the trypan blue imaging might not be sufficient for concluding. Other assays, such as MTT assay, should be included. 

The recombinant SLO and SPN were utilized in this study to demonstrate their roles in the depletion of ß-NAD+ in skeleton muscle cells. Nonetheless, a large part of the evidence was based on the supernatants from different M-type GAS (M1, M3, and M5). It is well-known that the virulence of different M-type GAS could differ dramatically. For example, M3 GAS, which has been known to have an inactivating mutation in rocA. Inactivating rocA expression resulted in the depression of virulence factors expression. Also, the virulence factors expression patterns of M1 and M3 strains are different (Horstmann et al. 2015. Infect Immun 83:1068-1077. https://doi.org/10.1128/IAI.02659-14).To utilize isogenic mutants of the same GAS strain will be much helpful.

Other comments

Line 19-20: the log phase culture supernatants were utilized for assay in this study. The expression of GAS secreted protein is growth-phase-dependent, whether supernatants from the exponential or stationary phase would have a higher level of SLO/SPN and be more suitable for these assays?

Supplementary Fig. 2: 3H10 and IgG should be indicated in the figure legend.